# Debonding Detection in Aluminum/Rigid Polyurethane Foam Composite Plates Using A_0_ Mode LAMB Wave EMATs

**DOI:** 10.3390/ma16072797

**Published:** 2023-03-31

**Authors:** Xin Yang, Jiang Xu, Shuchang Zhang, Jun Tu

**Affiliations:** 1School of Mechanical Science and Engineering, Huazhong University of Science and Technology, Wuhan 430074, China; 2School of Mechanical Engineering, Hubei University of Technology, Wuhan 430074, China

**Keywords:** aluminum/rigid polyurethane foam composite plates, debonding, A_0_ mode Lamb wave, EMAT

## Abstract

Aluminum/rigid polyurethane foam composite plates (ARCPs) are widely used for thermal insulation. The interface debonding generated during manufacturing degrades the thermal insulation performance of an ARCP. In this study, the debonding of an ARCP, a composite plate with a porous and damped layer of rigid polyurethane foam (RPUF), was detected using A_0_ mode Lamb wave electromagnetic acoustic transducers (EMATs). The low energy transmission coefficient at the interface caused by the large acoustic impedance difference between aluminum and RPUF made the detection difficult. Based on these structural characteristics, an A_0_ mode Lamb wave with large out-of-plane displacement was used to detect the debonding. EMATs are preferred for generating A_0_ mode Lamb waves due to their advantages of being noncontact, not requiring a coupling agent, and providing convenient detection. A finite element simulation model considering the damping of the RPUF layer, the damping of the PU film at the interface, and the bonding stiffness of the interface was established. The simulation results indicated that the Lamb wave energy in the aluminum plate transmits into the RPUF layer in small amounts. However, the transmitted energy rapidly attenuated and was not reflected into the aluminum plate, as the RPUF layer was thick and highly damped. Therefore, energy attenuation was evident and could be used to characterize the debonding. An approximately linear relationship between the amplitude of the received signals and the debonding length was obtained. Experiments were performed on an ARCP using EMATs, and the experimental results were in good agreement with the simulation results.

## 1. Introduction

Aluminum/rigid polyurethane foam composite plates (ARCPs) are widely used in thermal insulation for cold chain trucks, household appliances, transportation pipes [1], and building walls [2,3]. As shown in Figure 1a, an ARCP consists of a thin aluminum plate and a rigid polyurethane foam (RPUF) layer that is much thicker than the aluminum plate. A microscopic photo of an RPUF is shown in Figure 1b. It can be seen that the RPUF consists of numerous polyhedral pore cells, and the pore cells are connected to each other by a polyurethane (PU) matrix [4]. The pore cells are closed pore structures with an average diameter of about 200 µm. In addition, there is a film of PU at the interface of the ARCP, which is generated during foaming and acts as an adhesive between the upper and lower layers. Debonding may occur at the interface of an ARCP during manufacturing, which decreases the thermal insulation performance of the ARCP. Ultrasonic testing is widely used for detecting the interface debonding of composite plates [5]. Liu et al. [6] used an ultrasonic wave in resonant mode to detect the debonding of adhesive layers in three-layer plates and characterized debonding with the power spectrum amplitude of the resonant signal. Xu et al. [7] detected the debonding of concrete-filled steel tubes using piezoelectric transducers (PZTs). These scholars used bulk waves to detect the debonding of composite plates. However, bulk wave detection is time-consuming as only a limited area can be detected each time. A_0_ mode Lamb waves, with large out-of-plane displacements, which are more sensitive to the interface debonding, are widely used for the debonding detection of composite plates. Li et al. [8] used A_0_ mode Lamb waves to detect the debonding of steel/carbon fiber polymer bonded structures and obtained the relationship between the time of flight of the first arrival wave package and the length of debonding. Ghose et al. [9,10] investigated A_0_ mode Lamb wave propagation in an elastic/viscoelastic (steel–rubber) bilayer plate and its interaction with interfacial debonding. Sun et al. [11] used the electromagnetic-pulse-induced acoustic testing method to generate antisymmetric mode Lamb waves for the detection of debonding in plastic composite/metal adhesive bonding plates. Ng et al. [12] studied the interaction of A_0_ mode Lamb waves with debonding in complex structures made by composite laminates. Unlike the composite plates in the above studies, the RPUF layer in ARCPs is thick, porous, and highly damped. Compared with nonporous polymer, the damping of RPUFs comes from not only the viscoelastic damping of the PU matrix but also the porous structure of the RPUF [13]. The PU film at the interface mentioned earlier is also damped because of its viscoelasticity. Due to the large thickness and damping of the RPUF layer, the Lamb wave energy transmitted into the RPUF layer is not reflected into the aluminum plate layer. Therefore, the RPUF layer can be regarded as a porous damping material of semi-infinite thickness. These structural and material characteristics of ARCPs make this study different from previous studies. At present, few studies can be found for composite plates consisting of a metal plate and a porous polymer layer of semi-infinite thickness.

PZTs are a common choice for Lamb wave generation [8,14,15]. However, PZTs are inconvenient for automatic detection as coupling agents are required. Laser beams can also be used to generate Lamb waves, but the high cost and large size of the laser generation devices make field application difficult [16,17]. EMATs are extensively used to generate Lamb waves because of the advantages provided by requiring no coupling agent, being noncontact, and enabling convenient detection [18]. However, compared with PZTs, EMATs have low transduction efficiency and, therefore, have a lower signal-to-noise ratio [19,20,21]. At the same time, when EMATs are used to generate Lamb waves, the multimode and dispersion characteristics of Lamb waves affect the detection [22]. Many scholars have conducted research on this issue. Liu et al. [23] proposed a magnetic-concentrator-type EMAT configuration to avoid the problems caused by the multimode characteristics of Lamb waves. Sun et al. [24] enhanced A_0_ mode Lamb waves and suppressed S_0_ mode Lamb waves by optimizing the permanent magnet configuration. Kang et al. [25] enhanced the Lamb wave amplitude by optimizing the EMAT geometry parameters and avoided the multimode and dispersion problems of Lamb waves by choosing reasonable coil spacing and an EMAT working frequency. Guo et al. [26] designed an EMAT consisting of a racetrack coil and periodic permanent magnets that could selectively increase the purity and amplitude of A_0_ and S_0_ mode Lamb waves.

This paper proposes an effective method for the debonding detection of ARCPs, a composite plate with a porous and damped layer of RPUF. Due to the large difference in acoustic impedance between aluminum and RPUF, the energy transmission coefficient at the interface is small. Therefore, the A_0_ mode Lamb wave, which has large out-of-plane displacements, was selected for detection. To avoid the dispersion of Lamb waves and the superposition of multiple modes of Lamb wave signals, 1 MHz was selected as the detection frequency. Finite element simulation (FES) was employed to investigate the propagation of Lamb waves in the ARCP. Due to the specificity of the material and the structure of the ARCP, the properties of the bonding interface and the damping characteristics of the RPUF were considered in the simulation. Because the bonding stiffness of the interface influences the energy transmission and reflection coefficients of the interface, a spring constant was introduced to characterize the bonding stiffness of the interface. In addition, the simulation model considered the damping of the RPUF layer caused by the viscoelastic PU matrix and porous structure as well as the damping of the PU film at the interface because they lead to the energy attenuation of the Lamb waves. Finally, experiments were performed to validate the simulation results. This paper provides a reference for the detection of the interface debonding in the composite plates consisting of a metal plate and a porous polymer layer of semi-infinite thickness.

## 2. Theory Background

### 2.1. Mode Selection

When Lamb waves with large out-of-plane displacements propagate in an ARCP, more energy is transmitted into the RPUF layer and then is completely attenuated. Hence, the antisymmetric mode Lamb wave with larger out-of-plane displacements is considered to be more sensitive to the interface debonding of composite plates [8,9,10] and was chosen for detection.

Because the acoustic impedance of aluminum and RPUF differs by about 100 times and the bonding interface is imperfect, the transmission coefficient of Lamb waves at the interface is small. Moreover, the energy transmitted into the RPUF layer is not reflected into the aluminum plate. Therefore, the dispersion curve of an ARCP and an aluminum plate has little difference. The mode and frequency of the Lamb wave can be selected according to the dispersion curve of the aluminum plate. The dispersion curves of Lamb waves in a 1 mm aluminum plate calculated by the Dispersion Calculator [27] are shown in Figure 2. The following principles should be followed when selecting the frequency:(1)To avoid the superposition of multimode Lamb wave signals, the frequency should be lower than the cutoff frequency of the A_1_ mode Lamb wave.(2)The phase velocity and group velocity are equal at the point with a slope of zero on the group velocity dispersion curve, which means that the Lamb wave does not disperse. Therefore, the point with a slope closer to zero on the group velocity dispersion curve is preferred.

Based on the above principles, an A_0_ mode Lamb wave with a frequency of 1 MHz was chosen for the debonding detection of an ARCP. The wave structures of A_0_ mode and S_0_ mode Lamb waves at 1 MHz in a 1 mm aluminum plate are shown in Figure 3. It can be seen that A_0_ mode Lamb waves at 1 MHz are mainly based on out-of-plane displacements, while S_0_ mode Lamb waves at 1 MHz are mainly based on in-plane displacements.

### 2.2. Principle of the A_0_ Mode Lamb Wave EMAT

The A_0_ and S_0_ mode Lamb waves simultaneously exist in a 1 mm aluminum plate when the frequency is 1 MHz. Because the A_0_ mode Lamb wave was chosen for detection, an EMAT configuration that enhanced the amplitude of A_0_ mode Lamb waves and decreased the amplitude of S_0_ mode Lamb waves was needed. 

The working principles of EMAT are as follows: When alternating current is applied to the meander-line coil, the induced eddy current is generated in the aluminum plate. The bias magnetic field generated by the permanent magnets and the induced eddy current generates Lorentz force in the aluminum plate, as shown in Equation (1).
(1)$$F_{L}=J\times B$$
where $F_{L}$ is Lorentz force, J is the eddy current density, and B is the magnetic flux density.

A modified EMAT configuration that can increase the horizontal magnetic flux density in the eddy current area [19] is shown in Figure 4. According to Equation (1), for this EMAT configuration, the Lorentz force in the z-axis direction increases, which benefits the generation of A_0_ mode Lamb waves [28,29].

The center frequency of the meander-line coil is determined by the spacing interval of adjacent wires and its relationship can be described by [30]
(2)$$l=\frac{C_{p}}{2f}$$
where l is the spacing interval of adjacent wires of the meander-line coil, f is the center frequency of the meander-line coil, and $C_{p}$ is the phase velocity of A_0_ mode Lamb waves at the chosen frequency. In a 1 mm aluminum plate, the phase velocity of an A_0_ mode Lamb wave at 1 MHz is 2332 m/s according to the dispersion curve in Figure 2. The spacing interval of adjacent wires of the meander-line coil with a center frequency of 1 MHz was calculated to be 1.16 mm.

## 3. FES of the Propagation of A_0_ Mode Lamb Waves in an ARCP

### 3.1. FES Modeling

In order to study the propagation of Lamb waves in an ARCP, FES was carried out with COMSOL Multiphysics FES software. The two-dimensional (2D) geometric model of the ARCP used in the simulation is shown in Figure 5. The guided wave could not continuously propagate in the RPUF layer because the RPUF was highly damped, so the debonding could be simplified to the structure shown in Figure 5. The debonding length was set from 0 mm to 100 mm in steps of 20 mm. The geometric parameters of the EMAT are shown in Figure 6. EMAT was used in the model to generate and receive Lamb waves in the ARCP based on the coupling of the ‘magnetic fields’ with ‘solid mechanics.’ The current signal in the exciting coil was a five-cycle Hanning window modulated sinusoidal signal with a center frequency of 1 MHz. The material parameters are shown in Table 1. The ‘standard linear solid’ of the ‘viscoelasticity’ model is suitable for describing the viscoelasticity of polymers and was used to describe the viscoelastic damping of the PU matrix. The ‘loss factor damping’ in the ‘spring foundation (for domain)’ was used to describe the damping caused by the porous structure. ‘Viscous damping’ was added to the interface to characterize the damping of the PU film. The bonding stiffness of the interface affects the transmission and reflection of Lamb wave energy at the interface and is commonly described by the spring constant [31,32,33]. Therefore, a ‘spring constant’ was added and the spring constant was set to 1 × 10^13^ N/m^3^ [34].

### 3.2. Simulation Results

The A_0_ and S_0_ mode Lamb waves were both generated and propagated along the positive and negative directions of the x-axis in the aluminum plate as shown in Figure 7. The kinetic energy density distribution of the A_0_ mode Lamb wave propagating in the bonding region at 37.9 µs is shown in Figure 8. The kinetic energy density on Line 1 in Figure 8 is shown in Figure 9. As shown in Figure 8 and Figure 9, the energy of the A_0_ mode Lamb wave was mainly concentrated in the aluminum plate, and only a small amount of energy was transmitted into the RPUF layer. The ratio of kinetic energy density on the upper and lower sides of the interface was calculated to be 67.5. The low transmission coefficient at the interface was caused by the large acoustic impedance difference between aluminum and the RPUF layer and the interface bonding stiffness. The energy that leaked into the RPUF layer was rapidly attenuated and was not reflected into the aluminum plate, which was due to the large damping property of the film and the RPUF layer. The Lamb wave energy continuously attenuated when the Lamb wave propagated in the ARCP. Therefore, although the amount of transmitted energy was low, the energy attenuation was evident and could be used to characterize the debonding.

As the model used in the simulation is 2D, the current density in the receiving coil was utilized to characterize the amplitude of the A_0_ mode Lamb wave, as shown in Figure 10. The signals contained the passing signals of A_0_ mode and S_0_ mode Lamb waves and the superimposed signals of the first echo of S_0_ mode Lamb waves at the left and right boundaries. The passing signals of the A_0_ mode Lamb wave are shown in Figure 10b. The relationship between the peak-to-peak value of the passing signals of A_0_ mode Lamb waves in the simulation and the debonding length are shown in Figure 11. The results showed that the relationship between the amplitude of A_0_ mode Lamb wave signals and the debonding length was approximately linear. 

## 4. Experimental Verification

### 4.1. Experiment Setup

To verify the simulation results, experiments were conducted on an ARCP. The ARCP used for the experiments is shown in Figure 12. The size of the ARCP was 600 mm × 600 mm. The thickness of the aluminum plate was 1 mm and that of the RPUF layer was about 70 mm. The RPUF in the ARCP was prepared by the polymerization reaction of polyether polyols with isocyanates in the presence of various auxiliary agents. The ratio of isocyanate groups to hydroxyl groups in the isocyanate and polyether polyol was 1:1. All the reactants were mixed together and stirred well. Then, the mixture was poured onto the aluminum plate surrounded by a mold and left to foam sufficiently. The foaming agent in the additive helped to create the gas, and then bubble cores formed and grew until the foaming was completed. The ARCP was divided into 6 parts with the debonding length set from 0 mm to 100 mm in steps of 20 mm. The parameters of the EMAT were the same as those in the simulation. The magnets and coils are shown in Figure 13. The experiments were conducted using the electromagnetic ultrasonic guided wave instrument developed by our laboratory [35]. The sampling rate of the instrument was 50 MHz, the passband of its filter was 500 kHz~1.5 MHz, and the gain of the instrument was 70 dB. The experimental arrangement is shown in Figure 14. A 5-cycle sinusoidal signal with a peak value of 1 kV and a center frequency of 1 MHz was used as the exciting signal.

### 4.2. Results and Discussion

The signals received in the experiments were similar to the simulation signals, as shown in Figure 15. From Figure 15a, it can be observed that each group of experimental signals contained two wave packets. The first one was the initial pulse signal, and the second one was the passing signal of the A_0_ mode Lamb wave. The S_0_ mode Lamb wave signals were not observable because they were immersed in the noise due to their small amplitude. According to the A_0_ mode Lamb wave passing signals shown in Figure 15b, the amplitude of the A_0_ mode Lamb wave increased with the increase in the debonding length. The experimental and simulation signals were normalized using the amplitude of the signals of the ARCP without debonding. The normalization results are shown in Figure 16. The peak-to-peak value of the received A_0_ mode Lamb wave signals in the experiments had an approximately linear relationship with the debonding length. This suggested that the amplitude of the A_0_ mode Lamb wave could be used to quantitatively describe the debonding length. The experimental results and the simulation results were in good agreement, but small differences still existed. The differences could have been caused by errors in the damping parameters of the RPUF set in the simulation and the signal noise of the instrument. 

Sun et al. [11,18] found that the amplitude of the A_0_ mode Lamb wave could be used to characterize the debonding of double-layer plates but did not obtain the relationship curve between debonding length and the amplitude of the A_0_ mode Lamb wave. Ghose et al. [9] obtained a linear relationship between the debonding length and A_0_ mode Lamb wave amplitude in their study of steel/rubber double-layer plate debonding. Although the rubber layer in the composite plate studied also had viscoelasticity, it was not a porous material. So, there was no need to consider the damping caused by the porous structure. The proposed detection method could effectively detect the debonding in an ARCP and quantify the debonding length. The simulation model in this paper can provide a reference for the detection of debonding in other composite plates consisting of a metal plate and a porous polymer layer of semi-infinite thickness. We can also observe that the arrival time of the passing signals of the A_0_ mode Lamb wave could be related to the debonding length. However, due to the fast velocity of the A_0_ mode Lamb wave at 1 MHz and the relatively small debonding length set in the experiment, the differences in the arrival time of each group of experimental signals were small and would have been easily affected by the position error of the EMAT. Therefore, the arrival time of the A_0_ mode Lamb wave at 1 MHz could not be used to quantitatively characterize the debonding with a length less than 100 mm in the experiments.

## 5. Conclusions and Future Works

In this paper, an effective method for the debonding detection of an ARCP, a composite plate with a porous and damped layer of rigid polyurethane foam (RPUF), was proposed. The simulation results show that the energy transmission coefficient at the interface is small. The kinetic energy density of the A_0_ mode Lamb waves on both sides of the interface at t = 37.5 µs is up to 67.5 times according to the simulation. Moreover, the energy transmitted into the RPUF layer rapidly attenuates to zero in the RPUF layer and is not reflected into the aluminum plate because of the high damping of the thick RPUF layer. In addition, the energy continuously attenuates as the Lamb wave propagates in the ARCP. Therefore, the energy attenuation of the Lamb wave is large enough to characterize the debonding despite the low transmission coefficient at the interface. An approximately linear relationship between the amplitude of the A_0_ mode Lamb wave and the debonding length was obtained in the simulation. Experiments were conducted to verify the simulation, and the experimental results agreed well with the simulation results. The research results show that the A_0_ mode Lamb wave EMAT can be effectively used for the debonding detection of an ARCP and provide a reference for the interface debonding detection of composite plates consisting of a metal plate and a porous polymer layer of semi-infinite thickness.

In future work, the relationship between the arrival time of the A_0_ mode Lamb wave and debonding length in the ARCP will be studied. The case where multiple debonding regions exist on the Lamb wave propagation path at the same time will be considered. In addition, the influence of the debonding located below the EMAT on the Lamb wave generation and reception will be studied.

## Figures and Tables

**Figure 1 materials-16-02797-f001:**
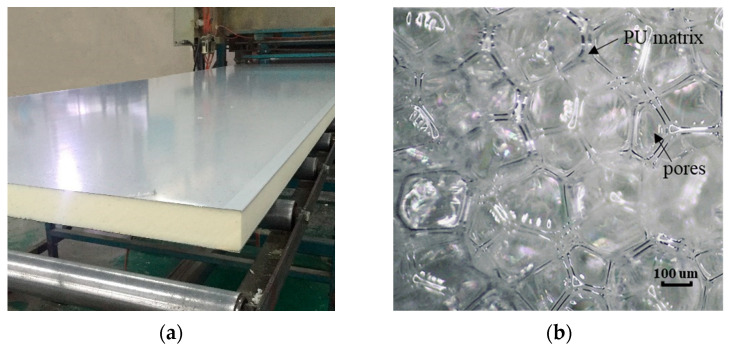
Photos of an ARCP and RPUF. (**a**) A photo of an ARCP; (**b**) a microscopic photo of RPUF.

**Figure 2 materials-16-02797-f002:**
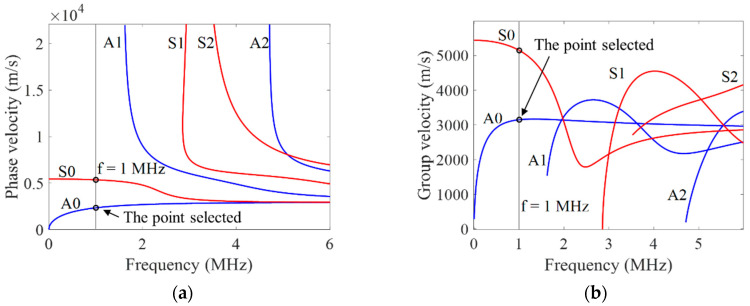
Lamb wave dispersion curves in 1 mm aluminum plate. (**a**) Phase velocity dispersion curves; (**b**) group velocity dispersion curves.

**Figure 3 materials-16-02797-f003:**
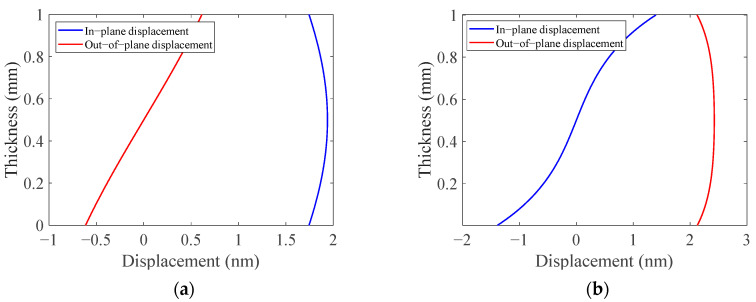
The wave structure of A_0_ and S_0_ mode Lamb waves at 1 MHz in 1mm aluminum plate. (**a**) S_0_ mode Lamb wave structure; (**b**) A_0_ mode Lamb wave structure.

**Figure 4 materials-16-02797-f004:**
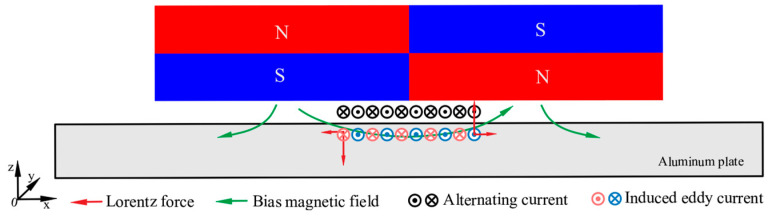
The configuration and principle of the A_0_ mode Lamb wave EMAT.

**Figure 5 materials-16-02797-f005:**
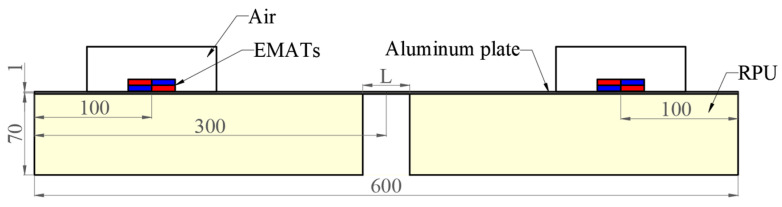
The 2D geometric models used in the simulation.

**Figure 6 materials-16-02797-f006:**
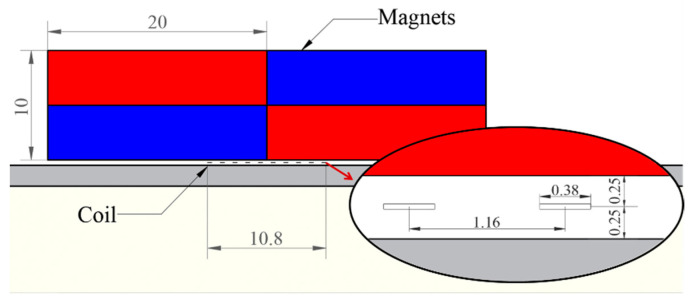
The geometric parameters of the EMAT.

**Figure 7 materials-16-02797-f007:**
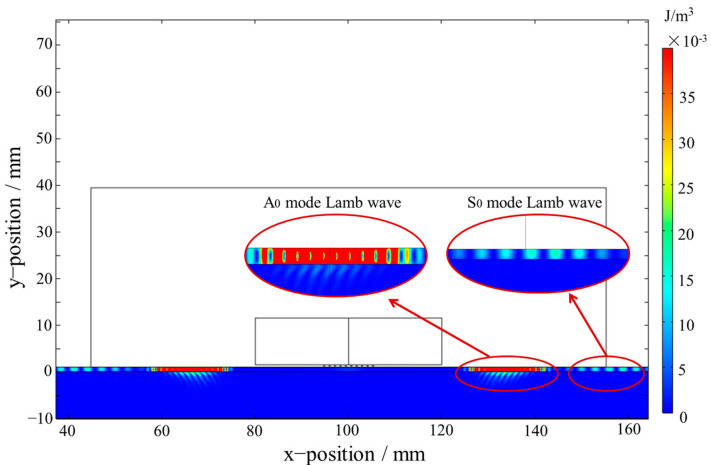
A_0_ and S_0_ mode Lamb waves in the ARCP.

**Figure 8 materials-16-02797-f008:**
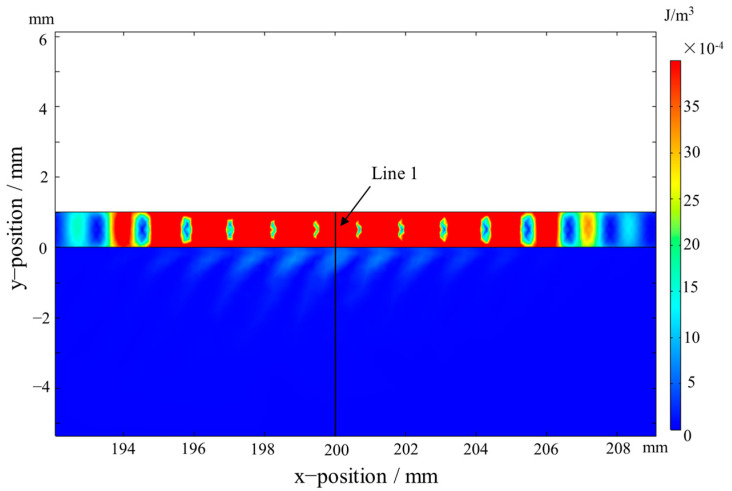
The kinetic energy density distribution of A_0_ mode Lamb waves in the bonding region.

**Figure 9 materials-16-02797-f009:**
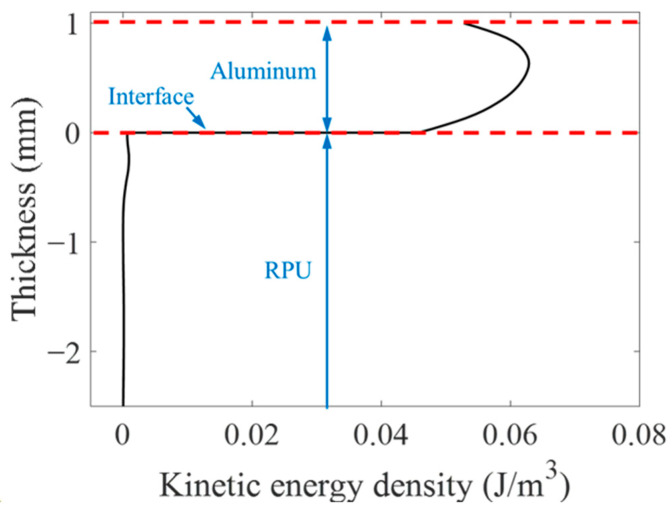
The kinetic energy density of A_0_ mode Lamb waves along the y-axis direction at Line 1. (Black line: the kinetic energy density; red line: interface; blue line: labels of interface and materials.)

**Figure 10 materials-16-02797-f010:**
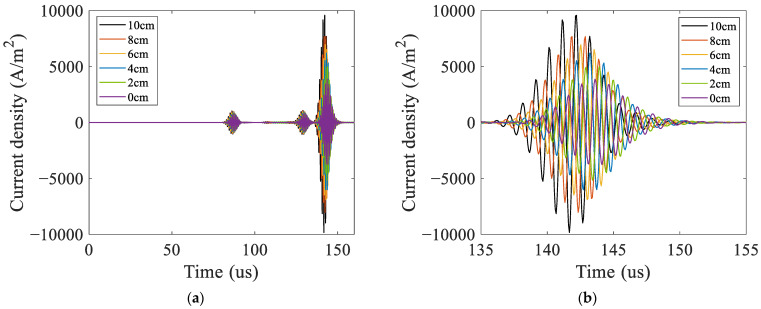
Received signals of the ARCP with debonding in simulation. (**a**) The current density received; (**b**) the passing signals of A_0_ mode Lamb waves.

**Figure 11 materials-16-02797-f011:**
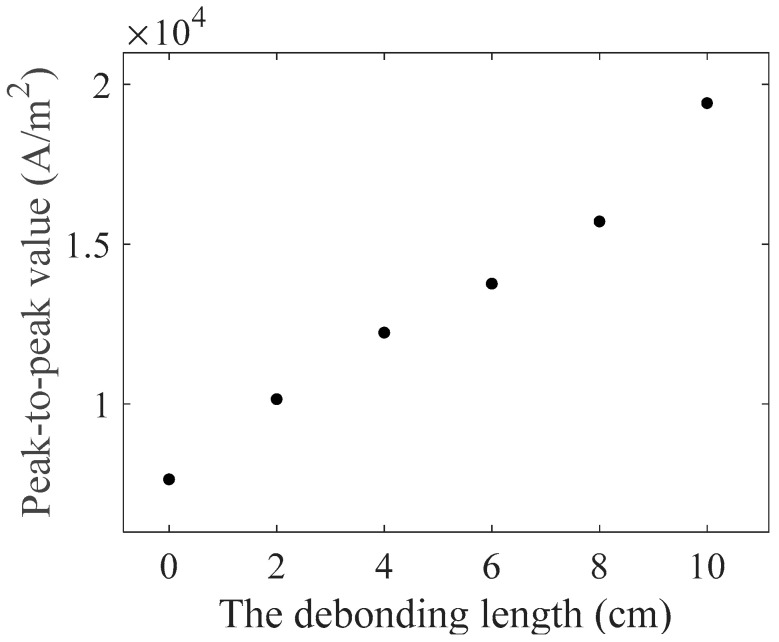
The relationship between the peak-to-peak value of A_0_ mode Lamb waves passing signals and the debonding length.

**Figure 12 materials-16-02797-f012:**
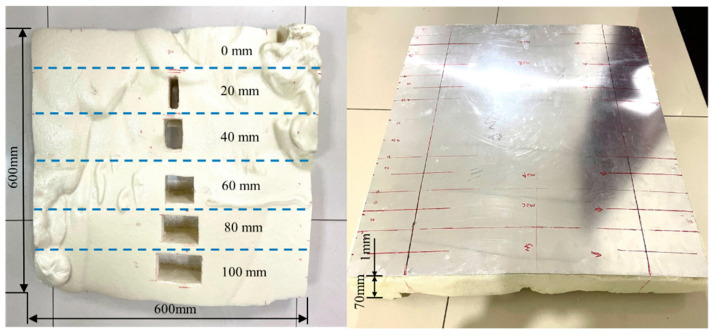
The ARCP used in the experiments.

**Figure 13 materials-16-02797-f013:**
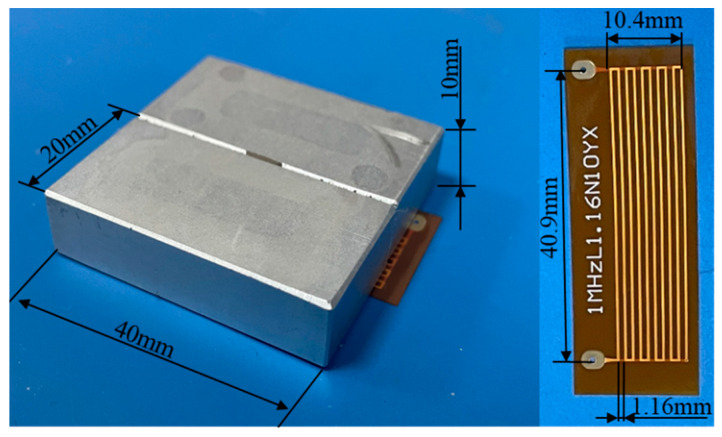
The magnets and coil of exciting and receiving EMAT.

**Figure 14 materials-16-02797-f014:**
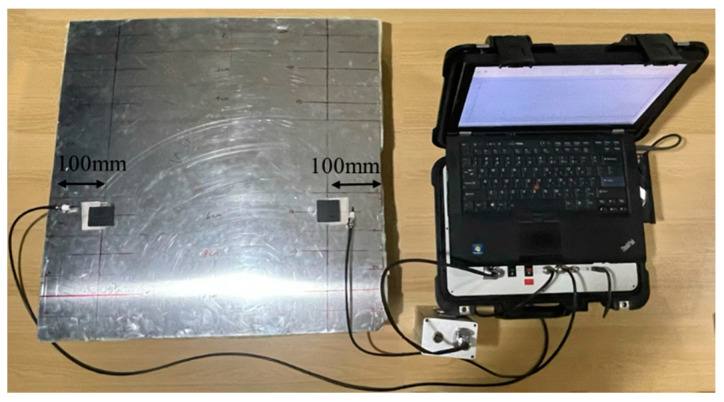
The photo of the general arrangement in the experiments.

**Figure 15 materials-16-02797-f015:**
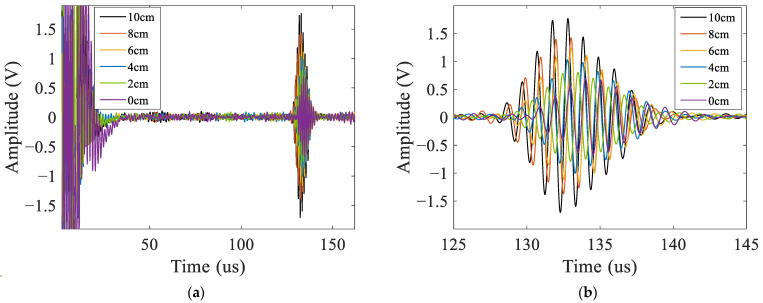
Signals received in the experiments. (**a**) All the signals received; (**b**) the passing signals of A_0_ mode Lamb waves.

**Figure 16 materials-16-02797-f016:**
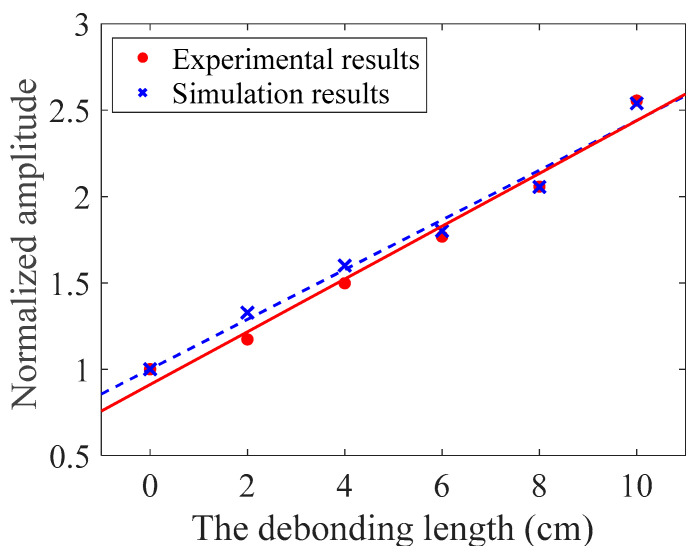
The relationship between the normalized amplitude of signals and the debonding length.

**Table 1 materials-16-02797-t001:** Parameters of materials.

Material	Density/kg·m^−3^	Young’s Modulus/GPa	Poisson’s Ratio
Aluminum	2700	70	0.33
RPUF	40	0.16	0.32

## Data Availability

The data supporting reported results by the authors can be sent by e-mail.

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
