# Peer review of "Debonding Detection in Aluminum/Rigid Polyurethane Foam Composite Plates Using A0 Mode LAMB Wave EMATs"

_materials, 2023, doi:10.3390/ma16072797_

Round 1

Reviewer 1 Report

The authors prepared aluminum/rigid polyurethane foam composite plates for debonding detection using A0 mode Lamb wave EMATs. The manuscript was prepared well and discussed the debonding detection by using electromagnetic acoustic transducers (EMAT) to generate A0 mode Lamb. This manuscript can be accepted after the minor revision.

1. Author should perform morphological characterization of the composite materials by SEM

2.  page 5, line 160, please check the error.

3. Please provide the x and y axis label for Figures 7 & 8

4. Author should add some more references from MDPI journals. 

Author Response

Thank you for your comments and suggestions, please check the attachment for the responses.

Reviewer 2 Report

1. English in Abstract, Introduction, Results and Conclusion should be improved.

2. Introduction should contain text describing the main aims of the paper and which tasks will be completed. The text about the obtained results has no place in Introduction section – this text should be transferred to Conclusions section.

3. Authors should correct the reference list according to journal requirements.

Author Response

Thank you for your comments and suggestions, please check the attachment for the response.

Reviewer 3 Report

The article is well-written and demonstrates the utility of the EMAT to detect the debonding in ARCP. Please recheck the references in the introduction.

Author Response

(The authors gave the same response as above.)

Reviewer 4 Report

Review of Debonding detection in aluminum/rigid polyurethane foam composite plates using A0 mode Lamb wave EMATs

Authors, in article showed a simulation model considering the damping of the RPUF layer caused by the viscoelastic RPU matrix and the porous structure, the damping of the interface, and interface bonding stiffness was established to study the propagation of the A0 mode Lamb wave in the ARCP.

1.      1. Why was aluminum chosen as the outer layer of the plate?

2.      2. What's new here since A0 mode Lamb wave propagation has already been used [6-11]. Given: ,,In this paper, the electromagnetic acoustic transducers (EMAT) for generating the A0 mode Lamb wave were employed to detect the debonding of ARCP. This requires more explanation in order to study how this method is better than the previous ones.

3.      Werse 56 : ,, ..few studies can be found for debonding detection of the metal/ semi-infinite porous polymer composite plates.’’. Does it make sense to produce such a board in which the layers peel off from each other?

4.      ,, semi-infinite porous polymer’’- what material is it? Please check if the word ,,infinite’’ has been translated correctly, it is incomprehensible.

5.      Does the thickness of the aluminum plate affect adhesion and insulation?

6.      Werse 97: author writed ,,acoustic impedance ‘’ but werse 29 ,,The ARCP are widely used in insulation’’. What kind of insulation are we talking about here, thermal, acoustic…?

7.      The authors used the latest bibliography, but in a small number of items.

8.      37.9 us ?

9.      which was due to the large damping property of the film and the RPUF layer. What kind of foil (film) are you talking about? Is it aluminum foil or was there some extra foil?

10.   4.1.Experiment Setup- I understand that the foam was not foamed in the form, but poured onto the sheet? And how was the thickness of the foam layer controlled during pouring? By what method was it foamed?

11.   There is no technology for receiving RCP into ARCP.

Author Response

(The authors gave the same response as above.)

Reviewer 5 Report

This manuscript is linguistically correct, I did not notice any editorial errors.

Unfortunately, the scientific level is very low, the whole text is based on the comparison of one research parameter.

The text in this form does not deserve publication in the Materials journal, which requires much more from the authors of the submitted papers.

My opinion: reject

Author Response

(The authors gave the same response as above.)

Reviewer 6 Report

Review of the manuscript ID – materials-2233848

Debonding detection in aluminum/rigid polyurethane foam composite plates using A0 mode Lamb wave EMATs

The manuscript submitted for review concerns the detection of debonding in aluminum/rigid polyurethane foam (ARCP) composite panels. This issue is important from the point of view of the efficiency of the manufacturing process, performance characteristics and the range of applications of this type of composite panels. The issue raised by the Authors is one of the important problems of plate composites, also often called sandwich.

In order to detect debonding between the cladding plate and the core (in this case, an aluminum plate and rigid PUR foam), the Authors of this paper proposed using an electromagnetic acoustic transducer (EMAT) generating Lamb wave in A0 mode. They performed a simulation based on the finite element method to study the propagation of Lamb wave at the interface of ARCP plate layers. The simulation results were compared with experimental results. Based on them, it was concluded that the method proposed by the Authors can be used to detect interlayer debonding of ARCP plates, as well as other types of sandwich composites. As can be seen from the Authors' declaration, the results presented provide a basis for continuing research in this area. 

Specific comments:

1.     The reviewed manuscript was edited with great care. In the Introduction, through appropriate references to the literature, the Authors have comprehensively outlined the research problem of debonding that occurs at the interface between cladding plates and the polymer matrix and the methods for detecting such defects. They have identified the advantages of EMAT, but I think it is important to clearly emphasize what is new in the work presented by the Authors.

2.     I have reservations about the lack of a clearly formulated aim of the paper, which should be presented in the Abstract and at the end of the Introduction.

3.     In my opinion, the method used was described in a clear and comprehensive manner. However, the characterization of the PUR foam itself (e.g., polyol/polyisocyanate ratio, growth time, etc.) and the description of the ARCP panels manufacturing process are probably unsatisfactory. To improve the quality of the manuscript, I suggest supplementing its content with this information.

4.     The results of simulations as well as experiments are adequately presented. The formulated conclusions clearly correlate with the obtained results and observations. It would be worthwhile to supplement the Results and Discussion with references to the literature and papers of other researchers.

5.     There are some editorial shortcomings in the paper, including an error with regard to the cited literature item on page 5, line 160.

Recommendation:

Considering the substantive level of the reviewed manuscript, its scientific and applied value, and minor editorial shortcomings, I recommend this manuscript for publication after minor revisions.

Author Response

(The authors gave the same response as above.)

Round 2

Reviewer 5 Report

Manuscript was significantly improoved - accept.